# Dietary *Saccharomyces cerevisiae boulardii* CNCM I-1079 Positively Affects Performance and Intestinal Ecosystem in Broilers during a *Campylobacter jejuni* Infection

**DOI:** 10.3390/microorganisms7120596

**Published:** 2019-11-21

**Authors:** Francesca Romana Massacci, Carmela Lovito, Silvia Tofani, Michele Tentellini, Domenica Anna Genovese, Alessia Arcangela Pia De Leo, Paola Papa, Chiara Francesca Magistrali, Elisabetta Manuali, Massimo Trabalza-Marinucci, Livia Moscati, Claudio Forte

**Affiliations:** 1Istituto Zooprofilattico Sperimentale dell’Umbria e delle Marche ‘Togo Rosati’, 06126 Perugia, Italy; francescaromana.massacci@gmail.com (F.R.M.); c.lovito@izsum.it (C.L.); silvia.tofani-esterno@izslt.it (S.T.); m.tentellini@izsum.it (M.T.); da.genovese@izsum.it (D.A.G.); a.deleo@izsum.it (A.A.P.D.L.); p.papa@izsum.it (P.P.); c.magistrali@izsum.it (C.F.M.); e.manuali@izsum.it (E.M.); l.moscati@izsum.it (L.M.); 2Department of Agricultural and Food Sciences, University of Bologna, 40127 Bologna, Italy; 3GABI, INRA, AgroParisTech, Université Paris-Saclay, 78350 Jouy-en-Josas, France; 4Istituto Zooprofilattico Sperimentale del Lazio e della Toscana ‘M. Aleandri’, 00178 Roma, Italy; 5Department of Veterinary Medicine, University of Perugia, 06126 Perugia, Italy; massimo.trabalza@unipg.it

**Keywords:** gut microbiota, histomorphology, probiotic, *Faecalibacterium prausnitzii*, live yeast

## Abstract

In poultry production, probiotics have shown promise to limit campylobacteriosis at the farm level, the most commonly reported zoonosis in Europe. The aim of this trial was to evaluate the effects of *Saccharomyces* supplementation in *Campylobacter jejuni* challenged chickens on performance and intestinal ecosystem. A total of 156 day old male Ross 308 chicks were assigned to a basal control diet (C) or to a *Saccharomyces cerevisiae boulardii* CNCM I-1079 supplemented diet (S). All the birds were orally challenged with *C. jejuni* on day (d) 21. Live weight and growth performance were evaluated on days 1, 21, 28 and 40. The histology of intestinal mucosa was analyzed and the gut microbiota composition was assessed by 16S rRNA. Performance throughout the trial as well as villi length and crypt depth were positively influenced by yeast supplementation. A higher abundance of operational taxonomic units (OTUs) annotated as *Lactobacillus*
*reuteri* and *Faecalibacterium prausnitzii* and a lower abundance of *Campylobacter* in fecal samples from S compared to the C group were reported. Supplementation with *Saccharomyces cerevisiae boulardii* can effectively modulate the intestinal ecosystem, leading to a higher abundance of beneficial microorganisms and modifying the intestinal mucosa architecture, with a subsequent improvement of the broilers’ growth performance.

## 1. Introduction

Since 2005, campylobacteriosis has been the most commonly reported zoonosis in humans in the EU [1]. There was a significantly increasing trend in the number of food-born infections EU/EEA at levels and at country level in half of the European Members States between 2008 and 2016, causing an increase in public health costs. In 2008, the European Food Safety Authority reported that *Campylobacter* was present in 86% of chicken carcasses throughout Europe [2]. A major source of infection is the consumption of undercooked chicken meat or the mishandling of raw contaminated chicken meat products [3]. The more efficient method for the reduction of human infection is to reduce prevalence and the excretion directly at the farm level.

Many studies have analyzed the effects of probiotics supplementation on animal performance, intestinal ecosystem and morphology, obtaining controversial results [4,5]. The beneficial effects of probiotics in animal production depend on a number of factors, including the chosen strain, the level of supplementation, the duration and the frequency of exposure to the probiotic and the physiological condition of the host [6]. *Saccharomyces cerevisiae* is a well known yeast used worldwide and since ancient times for different purposes, from baking to balance human gut microflora. This particular yeast species exerts probiotic influence in broilers by promoting the metabolic processes of digestion and nutrient utilization [6]. In recent years, studies about the dietary administration of this yeast in animal have increased [7,8,9]. Yet Line et al. [10] demonstrated that mannosyl residues present in yeast cell wall were able to bind harmful bacteria, proving that yeasts can be effective in pathogens regulation in the intestinal tract. To the best of our knowledge, no studies have considered the effect of *Saccharomyces cerevisiae boulardii* dietary supplementation against campylobacteriosis in poultry. Other studies [11,12,13,14] recorded a successful increase in productive performance in poultry, assessing an improvement in body weight (BW), feed intake (FI) and feed conversion ratio (FCR).

Beneficial effects can also be ascribed to modification in gut environment, including the morphology of the intestine. Muthusamy et al. [15] obtained positive modification of histomorphology of the intestinal mucosa of broilers with both live yeast and pellets of the yeast cell wall. Wang et al. [16] confirmed the beneficial activity of the live yeast, recording an improved villus surface area and villus width in the ileum of lipopolysaccharide-challenged broilers.

Moreover, probiotics become much more powerful and valuable when used as preventive health promoters and gut microbiota stabilizers [17]. There is an increasing amount of data suggesting that probiotics are able to maintain the eubiosis of the intestinal microbes by excluding and limiting the pathogens’ colonization and improving the abundance of beneficial bacteria in the gut [12,18,19,20,21,22,23,24]. Probiotic supplementation has the added benefit of allowing reduced antibiotic use in poultry and directly addressing consumer demand for a healthier residue-free meat product [25].

The specifics of interplay between dietary supplementation with *Saccharomyces cerevisiae boulardii*, live performances, histology, gut microbiota composition and consequently, the hypothesised protective effect against *Campylobacter* colonization in broilers, needs to be further investigated to validate the possible beneficial effects of this microorganism.

The hypothesis behind this study is that dietary supplementation with *Saccharomyces cerevisiae boulardii* CNCM I-1079 could positively affect the in vivo performances and the gut ecosystem in terms of histological traits and fecal and cecal microbiota composition of experimentally *Campylobacter jejuni* challenged broilers.

## 2. Materials and Methods

### 2.1. Animal Experimental Design

A total of 156 day old male Ross 308 chicks were randomly divided into 12 replicates (experimental units) of 13 birds each and assigned to one of the two experimental diets (6 replicates for each treatment). Animals were raised for 40 days (d) according to the Aviagen^®^ recommendations [26].

Chicks were allocated at the animal experimental facility of the Istituto Zooprofilattico Sperimentale dell’Umbria e delle Marche “Togo Rosati” (Perugia, Italy). The trial was authorized by the Italian Ministry of Health (33/2018-PR of 31 January 2018) in accordance with the European and Italian regulations (Directive 2010/63/EU, D.L. 26/2014). The experiment was carried out under the supervision of certified veterinarians. Chickens were vaccinated at hatchery against Newcastle disease, Marek disease, infectious bronchitis and coccidiosis, then placed in floor pens with wood shavings as litter, supplemental heat in the first period, plastic waterer, and feed ad libitum according to the experimental group. The temperature was monitored in order to ensure the same environmental conditions for all the replicates.

Feeds were formulated according to NRC (1994) and provided by PROGEO® Società Cooperativa Agricola (Italy). Animals were fed the starter formula from d1 to d21 and growing-finisher formula from d21 to d40. Feed formulation and chemical composition of the basal diet (starter and finisher) is reported in Appendix A. The dietary treatments were *(i)* basal control diet (C) and *(ii)* C supplemented with 1*10^9^ colony forming units (CFU)/kg of *Saccharomyces cerevisiae boulardii* CNCM I-1079 (Levucell^®^ SB, Lallemand Inc., Canada) (S). At d21, chicks were orally challenged (0.1 ml per bird) with a stock culture of 10^5^ CFU of *C. jejuni*/ml. The health and vitality status of the subjects were evaluated twice daily and at the end of the trial, a post-mortem exam was performed on all animals.

On days 1, 21, 28 and 40, all the birds were weighed and feed intake was recorded for the calculation of the average daily gain (ADG) and the feed conversion ratio (FCR). On days 28 and 40, five broilers from each replicate were sacrificed and samples were collected for further analyses.

### 2.2. Feed Analyses

Samples of the feeds were collected weekly during the trial, and the nutritional composition was determined. The composition of the basal diets (starter and finisher) is indicated in Appendix A. AOAC methods were applied for feed analyses. In particular, method 934.01 [27,28] was used to determine the dry matter content of feed, procedures 976.06, 920.39 and 942.05 were followed to determine crude protein, crude fat, and ash, respectively. The study of the neutral detergent fiber, acid detergent fiber, and lignin (sa) was performed according to Van Soest et al. [29] with slight modifications. Sodium sulphite instead of amylase was used in the neutral detergent fiber procedure. The AOAC method 985.35 [30] and the AOAC method 964.06 [31] were applied for the determination of calcium and phosphorous concentrations, respectively.

### 2.3. Experimental Challenge

All the birds were oral challenged with 0.1 ml of a saline solution containing 10^5^ CFU/ml of *C. jejuni* on day 21. The *C. jejuni* strain used in the challenge was isolated from a clinical case of Campylobacteriosis in broilers and then identified as *C. jejuni* using PCR [32]. The isolate was collected in the Veterinary Diagnostic Laboratory of the Istituto Zooprofilattico Sperimentale dell’Umbria e delle Marche “Togo Rosati” and stored at −80 °C until use.

For the preparation of the challenge solution, the strain was thawed and incubated for 48 hours at 42 °C on Karmali *Campylobacter* medium under microaerophilic conditions (10% CO_2_, 5% O_2_, 85% hydrogen). Then, using the McFarland scale and the optical density (OD) of a spectrophotometer, a bacterial concentration of 10^8^ CFU/ml was obtained, corresponding to the 0.5 McFarland and confirmed by 550 nm (OD550). Then, the suspension was diluted in sterile saline solution in order to obtain a 10^5^ CFU/ml suspension.

### 2.4. Microbiological Analyses

On day 21, fresh fecal samples were collected from each replicate in pool. An aliquot of 1 g was collected from each sample and diluted in tubes containing 9 ml of 0.9% sterile saline solution. Specimens were 10-fold serially diluted in order to obtain dilutions from 10^−1^ to 10^−10^ according to UNI EN ISO 6887–1:2017 [33]. According to UNI EN ISO 10272-2:2017 [34], Modified Charcoal Cefoperazone Deossicolate agar was used for the detection and enumeration of *Campylobacter* spp. Plates were incubated in microaerophilic conditions at 41.5 °C for 48 hours and the colonies were counted and then identified by PCR [32]. The same analysis was performed on the cecal content at the two sampling times of the trial (days 28 and 40).

At sampling (d28 and d40), enumeration of *Saccharomyces* spp. was performed according to ISO 21527-1:2008 [35] from cecal samples. Plates of Dichloran Rose Bengal Chloramphenicol Agar were incubated at 25 °C for 5 days. The results are expressed as log_10_ CFU/g.

### 2.5. Histo-Morphometrical Analyses

On day 40, segments of duodenum (duodenal loop) of approximately 2 cm in length from 36 subjects (3 for each replicate/group) were collected for histological and morphometric analyses. The samples were fixed in 10% neutral buffered formalin solution and routinely embedded in paraffin wax. Serial sections 3–4 µm-thick were stained with Hematoxilin and Eosin (HE) and periodic acid-shiff (PAS). The HE stained sections were used to evaluate the villi height (from the tip of the villus to the villus-crypt junction) and the crypts depths (from the base up to the crypt-villus transition region) as reported by Aliakbarpour et al. [36], while PAS stain was applied to assess the number of Goblet cells. For both HE- and PAS-stained sections, morphological and morphometrical analyses were performed in 5 randomly selected fields. Images were digitalized using the microscope Eclipse Ci-L (Nikon Corporation, Japan) using NIS-Elements Br-2 as software (v 5.10; Nikon). The villi/crypt (V/C) ratio was subsequently calculated and analyzed.

### 2.6. Gut Microbiota Sampling and DNA Analysis

Fecal samples were collected on days 21 and 40 by placing clean cardboard on the floor of each replicate group. Fresh fecal material was collected within 10 minutes from excretion and homogenized. From each fecal homogenate, two aliquots of 200 mg were collected for the microbiota analyses on days 21 and 40. On day 40, five animals from each replicate were slaughtered and cecal contents were pooled together and homogenized. From each cecal homogenate, two aliquots of 200 mg were collected for the microbiota analyses. All the samples were directly frozen in liquid nitrogen and kept at −80 °C until processing.

From each fecal and cecal aliquot, total DNA was extracted using the QIAamp PowerFecal DNA Kit (Qiagen, Valencia, CA, USA), according to the manufacturer’s instructions.

Microbial profiling was performed using high-throughput sequencing of the V3-V4 hypervariable region of the 16S rRNA gene (2x300 bp paired-end reads) on an Illumina MiSeq platform following the standard Illumina sequencing protocol and by using primers PCR1F_343 (5′-CTTTCCCTACACGACGCTCTTCCGATCTACGGRAGGCAGCAG-3′) and PCR1R_784 (5′-GGAGTTCAGACGTGTGCTCTTCCGATCTTACCAGGGTATCTAATCCT-3′).

### 2.7. Bioinformatic Analyses

The generated FastQ files were first quality-checked though the FastQC software (https://www.bioinformatics.babraham.ac.uk/projects/fastqc) and then analysed using the Quantitative Insights into Microbial Ecology pipeline (QIIME) v1.9.1 [37] by choosing the open-reference OTU calling approach [38]. The Illumina adapters were removed trough the “cutadapt” function [39]. Forward and reverse paired-end sequence reads were collapsed into a single continuous sequence according to the “fastq-join” option of the “join_paired_ends.py” command in QIIME. Therefore, the “split_libraries_fastq.py” command was used to demultiplex and filter (at Phred ≥ Q20) the fastq sequence data [40]. Subsequently, the sequences were clustered into an Operational Taxonomic Unit (OTU) against the GreenGenes database (release 2013-08: gg_13_8_otus) [41] by using the uclust algorithm [42] method at a 97% similarity cutoff. Chimeric sequences were removed through the “parallel_identify_chimeric_seqs.py” function in QIIME and by using the BLAST algorithm against the GreenGenes reference alignment [41]. Singleton OTU and OTUs with a number of sequences lower than 0.005% of the total number of sequences were removed from the analysis as recommended [43]. A phylogenic tree was generated from the filtered alignment using FastTree [44]. Any samples with less than 10000 post-quality control reads were removed from the analysis, which resulted in eliminating only one sample (fecal sample of group S on day 21).

### 2.8. Biostatistical Analyses

Performance data (BW, ADG, FCR) were analyzed through a repeated measure ANOVA procedure using the general linear model (GLM) procedure of SAS [45]. Sampling time and dietary treatment were included as fixed factors in the model and their interaction was evaluated using replicates as experimental units. Replicates were also included in the model to verify concordance within groups and showed no differences when nested within dietary treatment. All CFU data were transformed to respective log_10_ values before being analyzed with the same ANOVA model used for performance.

For the microbiota composition analysis, the biom OTU table was imported into R (v.3.6.0; www.r-project.org) using Phyloseq package (v.1.28.0) [46].

Feature abundance matrices were calculated at the OTU levels. The phyloseq [46], vegan (v.2.5-6) [47] and microbiome (v.1.6.0) [48] R packages were used for the detailed downstream analysis.

The α-diversity index was calculated with Shannon index using the phyloseq R package from OTU abundance table and richness was evaluated as the total number of OTUs present in each sample. The microbiome R package allowed us to study global indicators of the gut ecosystem state, including measures of evenness, dominance, divergences and rarity. All samples were normalized using the “rarefy_even_depth” function in the phyloseq R package, which is implemented as an ad hoc means to normalize microbiome counts that have resulted from libraries of widely differing size.

The β-diversity of microbiota composition between groups and time points was tested through the Whittaker’s index from OTU abundance table using the multivariate analyses of the homogeneity of group dispersion (the “betadisper” function of the vegan R package, that is, the distance of each replicates from the centroid of their dietary treatment).

Non-multiple dimensional scaling (NMDS) was used to visualize the β-diversity through the phyloseq and vegan R packages. The “envfit” function was implemented on the NMDS ordination matrix on days 21 and 40 separately, with Benjamin Hochberg’s multiple testing correction using the vegan R package. The significance was considered at an adjusted *p*-value < 0.05.

To assess whether the diversities across groups and time points were statistically different, an ANOVA using the “aov” function in R on α and β diversity and on log_10_ richness was performed followed by the post hoc Tukey’s HSD.

The permutational multivariate analysis of variance (PERMANOVA) using the Bray-Curtis distance was performed using the “adonis” function and the multiple-test correction of Benjamini-Hochberg (adjusted *p* < 0.05) in order to assess the community differences between groups.

A differential abundance analysis was performed using the function “fitFeatureModel” in the metagenomeSeq (v.1.26.3) R package at the OTU levels [49].

To characterize a set of microbes consistently present in the cecal microbiota of broilers, a OTUs detection threshold of 0.001% and a prevalence threshold of 99.9% (e.g., a given OTUs must be present in 99.9% of individuals in each time point with a relative abundance of at least 0.001%) was employed, using the microbiome R package [48].

## 3. Results

### 3.1. Animal Performances

Live weight was influenced by *Saccharomyces* supplementation throughout the trial (Table 1). The *Saccharomyces boulardii* supplemented group shower higher (*p* < 0.0001) BW on days 21, 28, and 40. ADG data followed the same trend at d21 and d28 (Table 1); at d40, no differences were highlighted between the groups. In the statistical model, in which the interaction between time and diet was included, FCR results were different only when the two fixed effects were considered alone (Table 1). Analyzing data and considering the dietary effect within each sampling time (Table 2), FCR results were lower in the S group on the first (days 0–21) and second (days 21–28) periods compared to the C group. At the final period (days 28–40), the FCR values of the two groups were comparable, following the same trend as that observed for ADG data.

Overall mortality data were comparable for the two experimental groups and within the range typical of the intensive poultry farming.

### 3.2. Microbiological Analyses

Microbiological analyses were performed in order to confirm the *Campylobacter jejuni* challenge and to evaluate the presence of the *Saccharomyces* spp. in feed.

Before challenge (day 21), all the birds in the two groups were culture-negative for *Campylobacter* spp. At sampling (days 28 and 40), *Campylobacter jejuni* was present in cecal content in all the replicates, thus confirming the success of the challenge (Appendix A).

*Saccharomyces* integration in feed was effective, with the yeast-supplemented group showing higher (*p* < 0.0001) yeast counts compared to the control group (Appendix A) in the cecal content analyses.

### 3.3. Histo-Morphometrical Analyses

The data obtained by examining intestinal morphometry at day 40 (Table 3) showed that supplemented birds had an increase in villi length (*p* < 0.0001) and crypt depth (*p =* 0.0333) compared to those from the control. No differences were observed in V/C ratio and PAS-positive cells count for both villi and crypt between the supplemented and control birds.

### 3.4. Gut Microbiota Analyses

Fecal (on days 21 and 40) and cecal (on day 40) samples were collected to profile the microbiota composition using 16S ribosomal RNA gene amplicon sequencing.

#### 3.4.1. Bacterial Phylogenetic Composition and Cecal Core Microbiota in Broilers

The raw sequencing data has been submitted to NCBI’s Sequence Read Archive (SRA) repository (BioProject: PRJNA578628; Biosample: SUB6446261, accessions 13154251 to 13154370). Before quality control, a mean of 82678.63 (S.D.= 22451.57) of the raw read count was available for each sample (Appendix A). After quality control, a mean of 58606.4 (S.D.= 16097.52) was obtained (Appendix A). OTU counts per sample and OTU taxonomical assignments are available in Appendix A. One sample on day 21 belonging to group S (replicate 3) was removed from the analysis for having less than < 10000 post-quality control counts. Sequences across the whole sample sets were successfully clustered into 1183 OTUs and only (1/1183) 0.08% of the OTUs could not be assigned to any phylum. Globally, 436 out of 1183 (37%) OTUs were annotated at the genus level.

Among the 1183 OTUs, the *Firmicutes* (1106/1183) and *Proteobacteria* (52/1183) phyla represented 93% and 4.4% of the annotated OTUs, respectively (Figure 1). Other phyla were present but with a lower percentage of OTUs (e.g., 1.2% *Tenericutes*, 0.40% *Cyanobacteria*, 0.25% *Bacteroides*, 0.25% *Actinobacteria*). The 33% (364/1106) of OTUs belonging the *Firmicutes* phylum was assigned to the *Lachnospiraceae* family, 26% (284/1106) to the *Ruminococcaceae* family and 9.1% (101/1106) to *Lactobacillaceae*. The 42% (22/52) of OTUs belonging the *Proteobacteria* phylum was assigned to the *Moraxellaceae* family, 40% (21/52) to the family *Enterobacteriaceae* and 12% (6/52) to *Campylobacteraceae*. The measures for each sample of evenness, dominance, divergences and rarity are reported in Appendix A.

To identify the stability of the microbiota, the core microbiota of each individual animal was investigated in our study (OTUs shared by 99% of samples with a minimum detection threshold of 0.001%). The cecal core microbiota in our cohort was composed by 26 OTUs, of which 96.15% (25/26) belonged to the *Firmicutes* phylum and 3.85% (1/26) to the *Tenericutes* phylum. The 96% (24/26) of OTUs belonging to the *Firmicutes* phylum was assigned to the *Clostridiales* order and more precisely, 38.4% (10/24) to the *Ruminococcaceae* family and 29.2% (7/24) to the *Lachnospiraceae* family (Figure 1C). The 4% (1/25) OTUs belonging to the *Firmicutes* phylum was assigned to the *Erysipelotrichales* order and could not be assigned to any family, similarly to for the OTU belonged to the *Tenericutes* phylum (Appendix A).

#### 3.4.2. Time Dependency of Fecal Microbiota Composition and Diversity

The overall composition of the fecal microbiota on days 21 and 40 (NMDS, Figure 2A,B) was driven by the dietary treatment (Adonis test: day 21, *p* = 0.00094; day 40, *p* = 0.00001). The beta diversity was not different between the experimental groups (ANOVA test, *p* > 0.05; Appendix A). In the NMDS plot, the centroids of the groups appeared separated, resulting in a non-significant *p*-value on day 21 (envfit test, *p* = 0.0547; Figure 2A) and a significant *p*-value at d40 (envfit test, *p* = 0.017; Figure 2B). Nevertheless, alpha diversity and the observed microbial richness at the OTU level did not show differences between the groups (ANOVA test, *p* > 0.05; Appendix A).

The dietary treatment was used in the model of the differential analysis at the OTUs level (Appendix A), describing, on day 21, nine differential abundance (DA) OTUs (Appendix A) and on days 40, 72 DA OTUs (Appendix A; Figure 2C). On day 21, the nine DA OTUs belonged to the order of *Clostridiales* and only three OTUs were assigned to the *Ruminococcaceae* family, being more abundant in the S group compared to the C group. On day 40, fecal samples were properly clustered according to the group of origin (Figure 2C) and the 72 DA OTUs were referred mainly to the family of *Lactobacillaceae*. Globally, OTUs belonging to the *Lactobacillus reuteri* and *Lactobacillus* spp. were described more abundantly in the S compared to the C group. From the other side, DA OTUs belonging to the *Lactobacillus agilis* and *Streptococcus* spp. were more abundant in the C group.

#### 3.4.3. Effects of Supplementation in Fecal Microbiota Composition and Diversity

When comparing each group at the two time points, we found that time significantly contributed to dissimilarities in the gut microbiota composition of the C and S groups (envfit test: C, *p* = 0.0001; S, *p* = 0.0001; Figure 3A,B). The PERMANOVA test also disclosed that the β-diversity of each group’s fecal microbiota changed over time (Adonis test: C, *p* = 0.00001; S, *p* = 0.00001). The beta diversity was different between the time points of the C group (ANOVA test: *p* = 0.0002; Figure 3C), whereas it was not different for the S group (ANOVA test: *p* > 0.05; Figure 3D). Alpha diversity showed differences between days 21 and 40 for both the C and S groups (ANOVA test: C, *p* = 0.001; S, *p* = 0.03; Figure 3C,D). In both groups, the observed microbial richness at OTU level did not show differences between the time points (ANOVA test: C, *p* > 0.05; S, *p* > 0.05; Figure 3C,D).

The dietary treatment was used in the model of the differential analysis at the OTUs level (Appendix A), describing among days 21 and 40, 298 DA OTUs in the C group (Appendix A) and 355 DA OTU in the S group (Appendix A; Figure 3E). In the C group, several DA OTUs were described as more abundant on day 21 compared with on day 40, such as *Oscillospira* and *Ruminococcus*. In the analysis of the S group, among the 355 DA OTUs, the genus *Lactobacillus* was found to be more abundant on day 40 when compared with the fecal composition of the S group on day 21. In both analyses, OTUs belonging to the genus of *Campylobacter* were differentially abundant, showing the absence of this bacterium on day 21 and confirming its presence on day 40. This finding confirms the success of the challenge.

#### 3.4.4. Time Dependency of Cecal Microbiota Composition and Diversity

The overall composition results of the cecal microbiota on day 40 (NMDS, Figure 4A) was linked to the dietary treatment (Adonis test, *p* = 0.00001). The beta diversity was not significantly different across the two groups (ANOVA test, *p* > 0.05). In the NMDS plot, the centroids of the two experimental groups appeared separated, resulting in a significant *p*-value (envfit test, *p* = 0.0001; Figure 4A). The alpha diversity at the OTU level and the observed microbial richness did not show differences among the groups (ANOVA test, *p* > 0.05; Figure 5B).

The differential abundance analyses show five DA OTUs on day 40 belonging mainly to the genus of *Lactobacillus* being more abundant in the S group when compared with the C group. One DA OTU belonged to the *Streptococcus* and was described to be more abundant in the C group (Appendix A).

#### 3.4.5. Abundances of OTUs Related to *F. prausnitzii* in Fecal and Cecal Contents According to Groups

There were five DA OTUs annotated as *F. prausnitzii* in the whole dataset (OTU IDs 589282, 157308, 158981, 158632 and New.ReferenceOTU11002). Since at least one of these OTUs was found DA in most of the comparisons between the groups and sampling time points, we decided to further explore the global abundance of *F. prausnitzii* by adding the abundances of the five OTUs in each sample. The normalized global abundance of *F. prausnitzii* on day 40 clearly shows an increase of abundance according to the dietary treatment, with the highest abundances observed in the S group, either in the fecal or cecal contents (Figure 5). The ANOVA analyses show differences both in fecal (*p* = 0.011) and cecal contents (*p* = 0.024) between the two groups.

#### 3.4.6. Supplementation-Related Effects on Abundances of OTUs Related to *Campylobacter* spp. in Fecal and Cecal Contents

There were five DA OTUs annotated as *Campylobacter spp.* in the whole dataset (OTU IDs 789621, 297260, 153941, 573432 and 1148151). Since at least one of these OTUs was found DA in most comparisons between groups and sampling time points, we decided to further explore the global abundance of *Campylobacter spp.* by adding the abundances of the five OTUs in each sample. The normalized global abundance of *Campylobacter spp.* on day 40 clearly shows a different abundance according to the dietary treatment, with the lowest abundances observed in the S group in the fecal samples and a similar abundance in cecal content among groups (Figure 6). On day 40, after the challenge performed on day 21, when animals resulted negative for *Campylobacter* spp. presence, ANOVA analyses showed significant differences in fecal samples (*p* = 0.0011), whereas in cecal contents, the abundance was not different between groups (*p* = 0.25).

## 4. Discussion

It is known that campylobacteriosis is an important worldwide public health problem with numerous socio-economic impacts [1]. There is also evidence that particular strains of *C. jejuni* and *C. coli* are aero-tolerant and hence, can survive in meat, as well as in the environment [50], showing the importance of limiting the spread of this bacterium at the farm level. In broilers, campylobacteriosis is an asymptomatic infection and *Campylobacter* species are found in farms and the surrounding environment.

Dietary administration of probiotics is generally reported as a positive factor able to influence animal productivity and health. The results obtained in the present study report that dietary supplementation with 1*10^9^ CFU/kg of *Saccharomyces cerevisiae boulardii* CNCM I-1079 is linked with an improvement in animal performance with a global beneficial effect on the gut microbiota composition and on intestinal mucosa, proving protective during a *Campylobacter jejuni* infection.

The use of probiotics in broilers is an increasingly common practise for the improvement of animal performances, the maintenance of a balanced gut microbiota, and the protection against the adverse effects of infectious diseases on these important production measures [4,18,51,52,53,54,55,56]. For example, Baidya et al. [57] showed that probiotics are the most effective growth promoters among all the zootechnical additives. However, the results of in vivo trials in chickens have been contradictory, some showing protection [8,9], and others, no effect [11,58].

Yeast supplementation is often reported to result in increased indices of animal performance [8,9,59]. In this study, the BW was higher in broilers fed a diet supplemented with *Saccharomyces cerevisiae boulardii*, exceeding the control group by more than 150 grams at day 40, thus confirming the positive impact of live yeast supplementation on broilers performances. Such a result can be important not only to confirm gut health improvement after *Saccharomyces* supplementation, but also to minimize economic losses due to *Campylobacter* infections.

The beneficial effects on performance we observed are in contrast with the results of a similar study in broilers using a similar supplementation protocol [14] that saw no effect on body weight, ADG or FCE. Despite the absence of significant zootechnical results, Fanelli et al. [14] reported modifications in intestinal and fecal microbiota, reducing *Salmonella* spp. enumeration in the neck skin, while no decrease on *Salmonella* spp. enumeration in faeces and cecal count. As for *Salmonella* spp., in their study, the *Campylobacter* colonization was lower in all the matrices examined when compared to the control group. In our study, no differences in cecal contents among groups according to the *Campylobacter jejuni* load were obtained. When the composition of the cecal microbiota of our cohort was investigated through the 16S rRNA sequencing approach, we did not record any differences according to the abundance of OTUs belonging to the genus *Campylobacter* between groups. On the contrary, in fecal samples, the genus *Campylobacter* showed a lower (*p* = 0.0011) abundance in the yeast supplemented group compared to the control group, suggesting that *Saccharomyces cerevisiae boulardii* supplementation could limit the excretion of *Campylobacter* in farm environment. It is known that *Campylobacter* appears to have a high ability to survive in the environment and hence, to colonize broilers, resulting in cross-contamination during slaughter and processing. Therefore, being able to limit the spread of this bacterium at the farm level should be one of the crucial goals to be achieved. It should be noted that although samples were not individually collected but homogenized in pool according to experimental replicate, this did not affect our data in term of the reliability of the results. In commercial farms, indeed, it is not easy to collect individual samples so investigations are often reported at the shed level, focusing more on environmental and group health status than single-bird evaluations; this is the scenario we wanted to reproduce and investigate in the present study.

Data obtained from performance indexes are supported in the present study by both histomorphometrical evaluation and microbiota composition analyses.

As for the architecture of intestinal mucosa, we obtained results comparable to other studies in which villi, crypts, or both were positively influenced by yeast supplementation in diets for broilers [60,61,62]. The higher number of PAS positive mucin producers cells is related in the literature to a protective effects for the intestinal mucosa [63]. In the present study, the absence of significant effects in their counts could be related to the intrinsic characteristics of the *Campylobacter*, which usually develop as an asymptomatic infection in poultry. Other factors to be considered are the infecting dose applied in the challenge and the methodologies used for the histological analyses, aimed at evaluating morphometrical changes more than the inflammation status of the mucosa. In any case, more in-depth study could and should be planned to investigate gene expression related to mucin production to better understand the mechanisms of action and variations related to dietary yeast supplementation.

The ability to balance the host gut microbiota with probiotics has been documented and the manipulation of the intestinal microbiota is considered a way to improve the health and growth performance of animals. Moreover, it is known that chicken performance is linked to the gut microbiota. A recent study showed that supplementation of *Saccharomyces* spp. in feed for broilers could modulate a healthier microbial ecosystem, subsequently enhancing the health status of broilers and improving the growth performance [22].

As reported in previous studies [64,65,66], we confirmed the role of the age of animals as a major influencer of the intestinal microbiota in broilers. In our study, we described a lower alpha diversity on day 40 compared to day 21, accordingly with the existing literature [64,65,66]. Moreover, we could speculate that the *Saccharomyces* supplementation could affect the diversity of the gut microbiota through time and following *C. jejuni* challenge. Indeed, on day 40, the control group showed a more heterogeneous gut microbiota composition when compared with the live yeast supplemented group, in which we reported a higher richness. In both the experimental groups, the *Ruminococcaceae* family was more abundant in young birds while the *Lactobacillus* genus was more abundant at the end of the trial [67,68].

Our results confirm *Firmicutes*, *Tenericutes* and *Proteobacteria* as major phyla observed in both fecal and cecal samples in all the groups and this is in accordance with different studies already present in literature [4,22,51,52,53]. An increase of *Lactobacillus* genus in yeast-supplemented animals in the fecal and cecal bacterial communities was observed in our study. At slaughter, yeast-supplementation clearly lead to an increased abundance of OTUs assigned to beneficial *Lactobacillus reuteri* and *Lactobacillus* spp., in agreement with another study [22] which found a higher abundance of *Lactobacillus* in the *Saccharomyces*-supplemented group compared to the control. Fecal and cecal bacterial communities of non-supplemented birds contained *Lactobacillus agilis* and *Streptococcus* spp. as the most abundant OTUs. Bacteria able to lower the pH of the gut by secreting lactic acid (e.g., *Lactobacillus* genus) are considered suitable candidates for pathogen control [4]. In fact, *Lactobacillus* strains are able to reduce the incidence of *Salmonella* spp., *Clostridium perfringens* and *Campylobacter* infections [4,69]. Moreover, *Lactobacillus* abundance is reported to be negatively correlated with *Campylobacter*, as confirmed in other studies where a decrease in *Lactobacillaceae* abundance was associated to increased levels of *Campylobacter* [70,71]. Several species and strains of *Lactobacillus*, including *L. reuteri*, *L. acidophilus*, *L. casei* and *L. rhamnosus*, have been extensively studied as probiotics supplementation in the prevention of human and animal diseases. *L. reuteri*, in particular, when supplemented in broilers’ diet, resulted in the enrichment of potentially beneficial lactobacilli and the suppression of *Proteobacteria*, including non-beneficial bacterial groups in the gut microbiota of broilers and increased carcass quality, decreasing pathogens’ contamination [72,73].

Another promising outcome of our results is represented by a higher abundance of OTUs annotated as *F. prausnitzii* in both the fecal and ceca content of the *Saccharomyces*-supplemented group compared to the control group, suggesting an increase of this potentially beneficial bacteria during the process of gut microbiota diversification. It is well established that *F. prausnitzii* plays an important role in GIT homeostasis, resulting in the reduction of enteric pathological status, becoming a gut health biomarker [74]. *F. prausnitzii* is considered one of the most promising next-generation probiotics (NGP) in humans due to its role in gut health with the amelioration of inflammation-related diseases [75]. In fact, *F. prausnitzii* decrease in the gut microbiota is correlated with enteric disorders in both humans and animals, such as Crohn’s disease, inflammatory bowel disease, colorectal cancer and irritable bowel syndrome [74,75,76,77,78,79,80,81,82,83,84,85]. A study showed a higher proportion of the genus *Faecalibacterium* in broilers with better FCR in comparison with those showing a low FCR; these data are in accordance with what we found in our study, in which the genus *Faecalibacterium* was more abundant in animals presenting lower FCR and consequently better performance [86,87,88].

## 5. Conclusions

Probiotic administration is recognized as one of the most promising dietary strategies to improve animal performance and welfare, reduce pathogen colonization and related adverse effects, and meet consumers’ demand for healthier and residue-free meat and meat products. Higher and deeper knowledge is needed to standardize probiotic use in livestock and the ability to counteract food borne pathogens. The more detailed investigations included in the present study involving not only the morphometrical evaluation of intestinal mucosa, but also the use of metagenomics analyses, could help in better understanding the effects of *Saccharomyces cerevisiae boulardii* CNCM I-1079 supplementation in broilers and the role of this probiotic in experimentally challenged animals.

In conclusion, the results of the present study revealed that *Saccharomyces cerevisiae boulardii* CNCM I-1079 supplementation could improve growth performance, effectively modulating the intestinal ecosystem, leading to a higher abundance of beneficial microorganisms and modifying the intestinal mucosa architecture, enhancing the health status of broilers and limiting the excretion of *Campylobacter*.

## Figures and Tables

**Figure 1 microorganisms-07-00596-f001:**
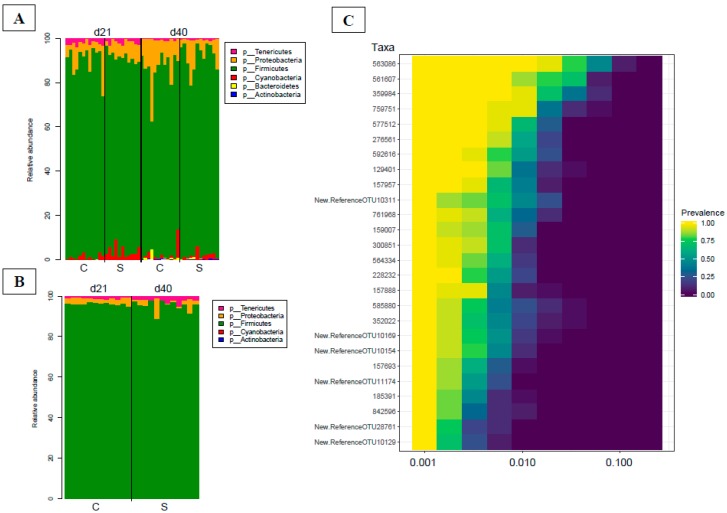
(**A**) Bar plot of the main phyla detected in fecal materials on days 21 and 40, respectively; (**B**) Bar plot of the main phyla detected in cecal materials on day 40; (**C**) Heatmap of the cecal core microbiota of broilers. OTUs were shared by 99% of individuals in our cohort, with a minimum detection threshold of 0.001%. The x-axis shows the detection threshold of the core microbiota operational taxonomic units (OTUs) in our cohort.

**Figure 2 microorganisms-07-00596-f002:**
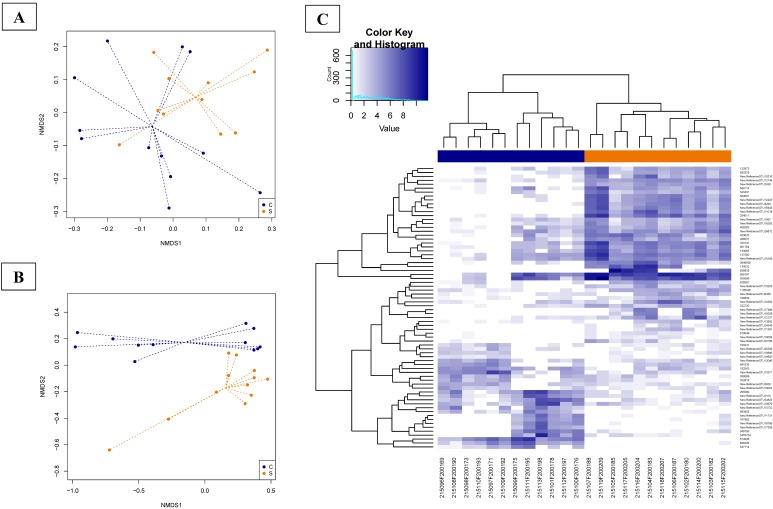
Dissimilarities in fecal microbiota composition represented by the non-metric multidimensional scaling (NMDS) ordination plot, with the Bray–Curtis dissimilarity index calculated on rarefied OTU abundances on days 21 (**A**) and 40 (**B**). The centroids of each group are features as the group name on the graph (“envfit”; vegan R package). The samples are colored by dietary treatment: C (basal control diet, blue) and S (basal diet supplemented with *Saccharomyces cerevisiae boulardii* CNCM I-1079; orange). The larger filled circles indicate group centroids. (**C**) Heat maps illustrating the abundances of differentially abundant (DA) OTUs expressed on day 40 among the fecal samples of basal control diet (C; blue) and the basal diet supplemented with *Saccharomyces cerevisiae boulardii* CNCM I-1079 (S; orange) groups.

**Figure 3 microorganisms-07-00596-f003:**
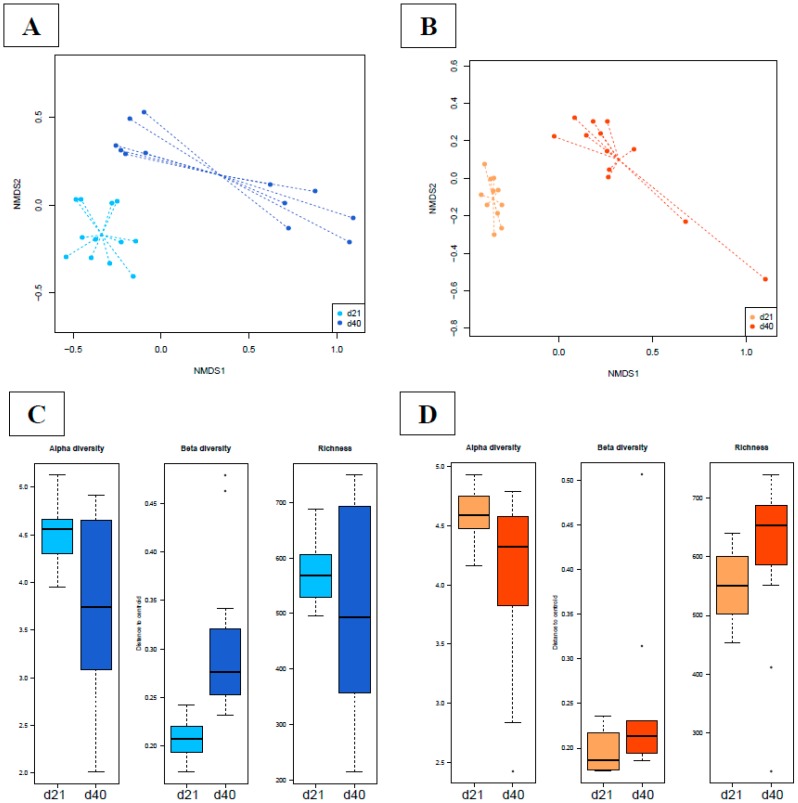
Dissimilarities in fecal microbiota composition represented by the NMDS ordination plot, using the Bray–Curtis dissimilarity index on OTU abundances among the C ((**A**): day 21, light blue; day 40, dark blue (**A**)) and S ((**B**): day 21, light orange; day 40, dark orange) groups. The centroids of each group are featured as the group name on the graph (“envfit”; vegan R package). The larger filled circles indicate group centroids. (**C**) Box plot representation of the alpha diversity using the Shannon index, beta diversity (Whittaker’s index) and richness using the rarefied OTUs of C (basal control diet) group at days 21 and 40. The samples are colored by time points: day 21 (light blue) and day 40 (dark blue). (**D**) Box plot representation of the alpha diversity (Shannon index), beta diversity (Whittaker’s index) and richness using the rarefied OTUs of S (basal diet supplemented with *Saccharomyces cerevisiae boulardii* CNCM I-1079) group at days 21 and 40; samples are colored by time points: day 21 (light orange) and day 40 (dark orange). (**E**) Heat maps illustrating the abundances of differentially abundant (DA) OTUs expressed on day 21 (light orange) and d40 (dark orange) among the fecal samples of the S group (basal diet supplemented with *Saccharomyces cerevisiae boulardii* CNCM I-1079).

**Figure 4 microorganisms-07-00596-f004:**
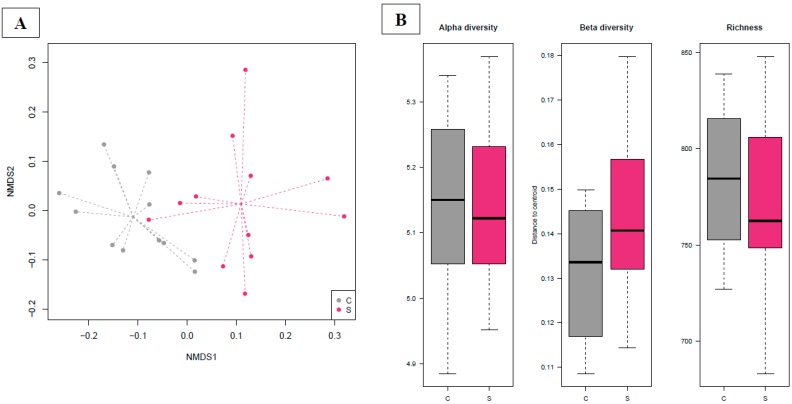
The figure includes only the samples obtained from cecal content. Dissimilarities in cecal microbiota composition represented by the non-metric multidimensional scaling (NMDS) ordination plot, with the Bray–Curtis dissimilarity index calculated on rarefied OTU abundances on day 40 (**A**). The centroids of each group are features as the group name on the graph (“envfit”; Vegan R package). Samples are colored by dietary treatment: C (basal control diet, grey) and S (basal diet supplemented with *Saccharomyces cerevisiae boulardii* CNCM I-1079; purple). The larger filled circles indicate group centroids. (**B**) Box plot graph representation of the alpha diversity (Shannon index), beta diversity (Whittaker’s index) and richness (total number of OTUs present in each sample) using the rarefied OTU table for each group on day 40; samples are colored by dietary treatment: C (basal control diet, grey) and S (basal diet supplemented with *Saccharomyces cerevisiae boulardii* CNCM I-1079; purple).

**Figure 5 microorganisms-07-00596-f005:**
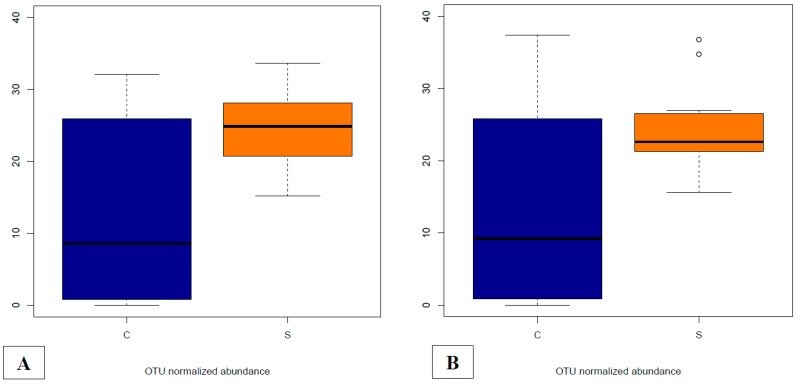
Abundances of *F. prausnitzii* on day 40 in fecal (**A**) and cecal (**B**) contents for each dietary treatment group (C = basal control diet, S = basal diet supplemented with *Saccharomyces cerevisiae boulardii* CNCM I-1079). Abundances were calculated as the addition of metagenomeSeq normalized for OTUs annotated as *F. prausnitzii* in the whole dataset (OTU IDs 589282, 157308, 158981, 158632 and New.ReferenceOTU11002).

**Figure 6 microorganisms-07-00596-f006:**
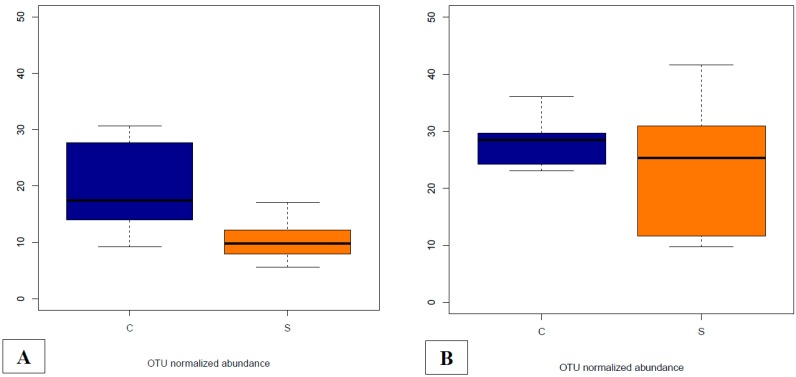
Abundances of *Campylobacter* spp. on day 40 in fecal (**A**) and cecal (**B**) contents for each dietary treatment group (C = basal control diet, S = basal diet supplemented with 1 × 10^9^ cfu/kg of *Saccharomyces cerevisiae boulardii* CNCM I-1079). The abundances were calculated as the addition of metagenomeSeq normalized for OTUs annotated as *Campylobacter* in the whole dataset (OTU IDs 789621, 297260, 153941, 573432 and 1148151).

**Table 1 microorganisms-07-00596-t001:** Effect of *Saccharomyces cerevisiae boulardii* CNCM I-1079 supplemented diets on the growth performance of chickens (g).

**Items**	**Diet x Time**					**Effects**	***p***
**Live weight**	**Day 1**	**Day 21**	**Day 28**	**Day 40**			
**C**	40.74 G	464.45 E	824.23 C	1791.46 B		**Diet**	<0.0001
**S**	36.14 G	667.11 F	1078.67 D	2021.03 A		**Time**	<0.0001
						**Diet × time**	<0.0001
**SEM**	17.980	17.980	17.980	24.24			
**ADG**		**Day** **0–21**	**Day** **21–27**	**Day** **28–40**	**Overall**		
**C**		21.18 D	59.96 B	74.66 A	51.94	**Diet**	0.0008
**S**		31.53 C	68.75 A	72.84 A	57.71	**Time**	<0.0001
						**Diet × time**	0.0062
**SEM**		1.901			1.098		

ADG: average daily gain. C: control diet. S: control diet supplemented with 1 × 10^9^ CFU/kg of *Saccharomyces cerevisiae boulardii* CNCM I-1079. SEM: standard error of the mean. A, B within item, means without a common letter differ (*p* < 0.05).

**Table 2 microorganisms-07-00596-t002:** Effect of *Saccharomyces cerevisiae boulardii* CNCM I-1079 supplemented diets on feed conversion ratio (FCR) analyzed by trial period.

FCR	Day 0–21	Day 22–27	Day 28–40
**C**	1.87 A	1.95 A	2.08
**S**	1.72 B	1.85 B	1.92
**SEM**	0.046	0.050	0.061
***p***	0.0423	0.024	0.0837

C: control diet. S: control diet supplemented with 1*10^9^ CFU/kg of *Saccharomyces cerevisiae boulardii* CNCM I-1079. SEM: standard error of the mean. A, B within column, means without a common letter differ (*p* < 0.05).

**Table 3 microorganisms-07-00596-t003:** Effect of *Saccharomyces cerevisiae boulardii* CNCM I-1079 supplemented diet on intestinal histology and morphometry.

Group	Morphometrical Analyses			PAS positive cells	
	Villi length (µ)	Crypt depth (µ)	V/C ratio	Villi	Crypt
**C**	2688.92 B	352.66 B	7.965	136.99	19.48
**S**	3870.82 A	428.78 A	9.714	188.18	20.76
**SEM**	111.844	20.504	0.540	27.748	0.808
***p***	<0.0001	0.0333	0.0617	0.1943	0.265

SEM: standard error of the mean. C: control diet. S: control diet supplemented with 1*10^9^ CFU/kg of *Saccharomyces cerevisiae boulardii* CNCM I-1079. A, B within column, means without a common superscript differ (*p* < 0.05).

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
