# Peer review of "Dietary Saccharomyces cerevisiae boulardii CNCM I-1079 Positively Affects Performance and Intestinal Ecosystem in Broilers during a Campylobacter jejuni Infection"

_microorganisms, 2019, doi:10.3390/microorganisms7120596_

Round 1

Reviewer 1 Report

The authors challenged with C.jejuni two groups of animals fed with S.cerevisiae supplemented and not supplemented diet. They then evaluated several parameters including body weight, feed intake, histological and morphometric parameters among others and microbiome composition during the time (two sampling points).

The manuscript is well written and methodology is rigorous and well described, although it uses an obsolete pipeline. Nevertheless, the methodology they applied is sophisticated and well performed.

Results are, on average well exposed but, in my opinion, some adjustments are necessary. 

They demonstrated the positive effects of S.cerevisiae supplementation in terms of body weight, feed intake and histological parameters but changes in microbiome composition (which have been well analyzed and reported)  did not support the title of the manuscript that claims: "positively affects performance and intestinal ecosystem... during a campylobacter jejuni infection".

first of all, the "performance" has to be specified and quantified. It is not clear if there is any performance indicator... then, it is not clear how this supplemented diet can affect the performance...

Moreover, they claim on page 13 "..in cecal contents the abundance was not different among groups." while they are referring to the amount of Campylobacter abundance in the cecal content; this suggests to me that Campylobacter can be considered in the "resident" microflora. 

I perfectly understand (figure 6) that the data suggest a reduction of the campylobacter load in the cecal content due to the diet supplementation but both data dispersion and statistical analysis provide just an indication.

The effects of S.cerevisiae are not new and are well documented, anyway, the data here proposed are interesting and well structured. I suggest some revision in data reporting and, if possible, a more detailed description of differences in the gut communities.  

page 1, lines 38-39 authors claimed: "obtaining contradictory results", it's fine but references report both positive effects, please include references with "no positive" results or better specify.

page 1. line 43 ..human gut microflora. Reference required.

page2 line 21. major concern: please, define what is meant by "...affects performances... in terms of ... and gut microbiota composition..."

page 4. paragraph 2.7 minor concern. although the methodology here proposed is fundamentally correct and well exposed, the QIIME pipeline has been discontinued in dec.2017, then substituted by the QIIME2. The authors should consider the use of this latter, also considering the low amount of OTUs classified at the genus level (36% in the results section). 

page 4. lines 19-20. the verb is lacking in this sentence

page 4 line 38. ...rarity. Please, specify which algorithms. 

page 8 line 1. the mean is given, please indicate dispersion. better to add a table indicating how many reads for sample.

page 8. line 22. please indicate the rationale behind the study of the "core microbiome"

page 9. Figure 2 panel C. major concern. Although graphically appealing, I believe that the heatmap should contain all OTUs; in this manner, it will provide a meaningful overview of the dataset.

page 9. Figure 2 panel C. minor concern. please increase the font size at the bottom side of figure

page 9. Figure 2 panels A, B. minor concern. Again, although graphically appealing dashes may drive the reader to mis-interpreter the data. In panel A, forgetting for an instant the colors, data will appear not particularly clustered... similarly in the panel B, data will form two well-separated clusters (one in the left, and one in the right). It could be interesting to investigate the reasons for such behavior.

page 10. Figure 3. Panel A. similar to the previous comment. Data clearly form three clusters; in particular, the group with the supplemented diet is formed by two well-separated clusters. This should be investigated.

page 11. Figure 3 panel E. see comment page 9. Figure 2 panel C.pa

page 12 paragraph 3.4.5 please indicate the rationale behind the investigation of the F. prausnitzii related OTUs

Author Response

Dear Editor,

I, on behalf of all the co-authors, thank you and the reviewers for the time and expertise invested in our manuscript “Dietary Saccharomyces cerevisiae boulardii CNCM I-1079 positively affects performance and intestinal ecosystem in broilers during a Campylobacter jejuni infection” (ID microorganisms-642415). We are sure that the amended version better frames our study for readers. We have modified the manuscript according to the comments and the suggestions of reviewers. The changes are clearly highlighted, in the text as required

Please, find below a point-by-point response to the reviewers along with the revised manuscript.

We hope you will find our revised manuscript improved and suitable for publication in Microorganisms.

Thank you again for your consideration.

Sincerely,

Claudio Forte

Reviewer 1

The authors challenged with C. jejuni two groups of animals fed with S.cerevisiae supplemented and not supplemented diet. They then evaluated several parameters including body weight, feed intake, histological and morphometric parameters among others and microbiome composition during the time (two sampling points).

The manuscript is well written and methodology is rigorous and well described, although it uses an obsolete pipeline. Nevertheless, the methodology they applied is sophisticated and well performed.

Results are, on average well exposed but, in my opinion, some adjustments are necessary.

They demonstrated the positive effects of S. cerevisiae supplementation in terms of body weight, feed intake and histological parameters but changes in microbiome composition (which have been well analyzed and reported) did not support the title of the manuscript that claims: "positively affects performance and intestinal ecosystem... during a campylobacter jejuni infection".

First of all, the "performance" has to be specified and quantified. It is not clear if there is any performance indicator... then, it is not clear how this supplemented diet can affect the performance...

AU: Performance are included in the text in terms of Live Weight, Average Daily Gain and Feed Conversion Ratio (Tables 1 and 2). These are in our opinion the most common performance indicators used in both experimental trial and in field activities. No performance at slaughter, in terms of meat quality and percentage yield were not included as soon as slaughter was not performed in a commercial slaughterhouse but at the experimental facility.

Moreover, they claim on page 13 "..in cecal contents the abundance was not different among groups." while they are referring to the amount of Campylobacter abundance in the cecal content; this suggests to me that Campylobacter can be considered in the "resident" microflora.

AU: Thank for the advice, we re-phrased the whole period to result more clear for the readers (Page 7, lines 29-30).

I perfectly understand (figure 6) that the data suggest a reduction of the campylobacter load in the cecal content due to the diet supplementation but both data dispersion and statistical analysis provide just an indication.

AU: We totally agree with the reviewer, data recorded showed a significant effect in faecal samples, not confirmed by cecal analyses. We have discussed the data analysing the possibility to limit the excretion of the Campylobacter in the environment (Page 9, lines 4-12).

The effects of S.cerevisiae are not new and are well documented, anyway, the data here proposed are interesting and well structured. I suggest some revision in data reporting and, if possible, a more detailed description of differences in the gut communities. 

page 1, lines 38-39 authors claimed: "obtaining contradictory results", it's fine but references report both positive effects, please include references with "no positive" results or better specify.

AU: modified

page 1. line 43 ..human gut microflora. Reference required.

AU: added

page2 line 21. major concern: please, define what is meant by "...affects performances... in terms of ... and gut microbiota composition..."

AU: We modified the text, making more precise the hypothesis behind our study.

page 4. paragraph 2.7 minor concern. although the methodology here proposed is fundamentally correct and well exposed, the QIIME pipeline has been discontinued in dec.2017, then substituted by the QIIME2. The authors should consider the use of this latter, also considering the low amount of OTUs classified at the genus level (36% in the results section).

AU: we agree with the reviewer that QIIME2 is a more updated pipeline for the 16S analyses, and indeed we are transitioning our pipelines towards Qiime2. However, given that the time agreed by the Editor for answering to the review is 5 days, we will not be able to reanalyse all data again with Qiime2. Nevertheless, in our experience using both of the pipelines (QIIME 1.9.1 and QIIME 2) in other datasets, we did not observe substantial changes in the conclusions of the analysis. We are evaluating how the amplicon sequencing variant (ASV) philosophy in Qiime2 could change some conclusions when compared to the OTU approach.  As regards to the taxonomical classification, using the GreenGene database is quite common to have these percentage of OTUs classified at the genera level, and it is not far from other previous studies in chickens and other animal species.

page 4. lines 19-20. the verb is lacking in this sentence

AU: We apologize for the mistake and we modified the text accordingly.

page 4 line 38. ...rarity. Please, specify which algorithms.

AU: We based our analysis on the standard indices that the function provides in the microbiome R package. The following rarity indices are provided: log_modulo_skewness, low_abundance, noncore_abundance and rare_abundance.

page 8 line 1. the mean is given, please indicate dispersion. better to add a table indicating how many reads for sample.

AU: We thank the reviewer for the comment and we deepened our information. We included the standard deviation of our data before and after the quality control of the read counts and we added a new table (table S3).

page 8. line 22. please indicate the rationale behind the study of the "core microbiome"

AU: added

page 9. Figure 2 panel C. major concern. Although graphically appealing, I believe that the heatmap should contain all OTUs; in this manner, it will provide a meaningful overview of the dataset.

AU: we totally agree with the reviewer and we have added in the supplementary materials the suggested heatmaps. Since the heatmap is created taking into account the differential analysis at the OTU level, we believe that it could be more representative of the OTUs differentially abundant between the two groups, clearly showing two different clusters.

page 9. Figure 2 panel C. minor concern. please increase the font size at the bottom side of figure

AU: We have increase the font size of the bottom side of the panel C as suggested.

page 9. Figure 2 panels A, B. minor concern. Again, although graphically appealing dashes may drive the reader to mis-interpreter the data. In panel A, forgetting for an instant the colors, data will appear not particularly clustered... similarly in the panel B, data will form two well-separated clusters (one in the left, and one in the right). It could be interesting to investigate the reasons for such behavior.

AU: we totally agree with the reviewer, in fact the beta diversity and the envfit analysis were not significant between groups in panel A. However, in order to avoid a mis-interpretation of the data we modified the page 3 line 10. As regards to the panel B, the beta diversity calculated within the two groups was not significant explaining an higher heterogeneity of the group.  

page 10. Figure 3. Panel A. similar to the previous comment. Data clearly form three clusters; in particular, the group with the supplemented diet is formed by two well-separated clusters. This should be investigated.

AU: we totally agree with the reviewer. More in detail, the beta diversity was different between groups in panel A, causing an higher heterogeneity of the group. We checked the NMDS ordination plot in order to find possible correlation between the three clusters as requested by the reviewer, and we did not find any particular issue. One possible explanation is linked to the beta diversity recorded values; in fact, it is known how the beta diversity increases with the age and this was in accordance with our data.

page 11. Figure 3 panel E. see comment page 9. Figure 2 panel C.pa

AU: we totally agree with the reviewer and we have added in the supplementary materials the suggested heatmaps. Since the heatmap is created taking into account the differential analysis at the OTU level, we believe that it could be more representative of the OTUs differentially abundant between the two groups, clearly showing two different clusters.  We have increase the font size of the bottom side of the figure 3 panel E as suggested.

page 12 paragraph 3.4.5 please indicate the rationale behind the investigation of the F. prausnitzii related OTUs

AU: The rationale behind the study of F. prausnitzii is deepened in the discussion section, in order to avoid to load heavily the results paragraph. Moreover, in the discussion, the importance of this bacterium for the gut homeostasis has been defined.

Reviewer 2 Report

In this manuscript, Romana et al. present an interesting study on how dietary supplementation with Saccharomyces cerevisiae boulardii CNCM I-1079 affects performance and intestinal ecosystem in broilers during a Campylobacter jejuni infection. The study introduces clear and sound, worth of considering to be published in Microorganisms

There is one issue which the authors need to address: the rationale for using Saccharomyces cerevisiae boulardii in this study, and not other S. c. varieties; this needs to be explained in the Introduction.

Another minor issue: pages 1-2, lines 45-1. Please replace "mannose molecules" with "mannosyl residues".

Author Response

Dear Editor,

I, on behalf of all the co-authors, thank you and the reviewers for the time and expertise invested in our manuscript “Dietary Saccharomyces cerevisiae boulardii CNCM I-1079 positively affects performance and intestinal ecosystem in broilers during a Campylobacter jejuni infection” (ID microorganisms-642415). We are sure that the amended version better frames our study for readers. We have modified the manuscript according to the comments and the suggestions of reviewers. The changes are clearly highlighted, in the text as required.

Please, find below a point-by-point response to the reviewers along with the revised manuscript.

We hope you will find our revised manuscript improved and suitable for publication in Microorganisms.

Thank you again for your consideration.

Sincerely,

Claudio Forte

-------

Reviewer 2

In this manuscript, Romana et al. present an interesting study on how dietary supplementation with Saccharomyces cerevisiae boulardii CNCM I-1079 affects performance and intestinal ecosystem in broilers during a Campylobacter jejuni infection. The study introduces clear and sound, worth of considering to be published in Microorganisms.

There is one issue, which the authors need to address: the rationale for using Saccharomyces cerevisiae boulardii in this study, and not other S. c. varieties; this needs to be explained in the Introduction.

AU: We thank the reviewer. We better explained the rationale for using Saccharomyces cerevisiae boulardii in our study (Page 2, lines 2-4 and 18-21)

Another minor issue: pages 1-2, lines 45-1. Please replace "mannose molecules" with "mannosyl residues".

AU: modified